# Impact of Fermentation Pretreatment on Drying Behaviour and Antioxidant Attributes of Broccoli Waste Powdered Ingredients

**DOI:** 10.3390/foods12193526

**Published:** 2023-09-22

**Authors:** Claudia Bas-Bellver, Cristina Barrera, Noelia Betoret, Lucía Seguí

**Affiliations:** Institute of Food Engineering-FoodUPV, Universitat Politècnica de València, Camino de Vera, s/n, 46022 Valencia, Spain; clbabel@etsiamn.upv.es (C.B.-B.); noebeval@tal.upv.es (N.B.); lusegil@upvnet.upv.es (L.S.)

**Keywords:** broccoli stems, waste valorisation, fermentation, drying, probiotic, functional powders

## Abstract

Valorisation of fruit and vegetable wastes by transforming residues and discards into functional powdered ingredients has gained interest in recent years. Moreover, fermentation has been recalled as an ancient technology available to increase the nutritional value of foods. In the present work, the impact of pretreatments (disruption and fermentation) on drying kinetics and functional properties of powdered broccoli stems was studied. Broccoli stems fermented with *Lactiplantibacillus plantarum* and non-fermented broccoli stems were freeze-dried and air-dried at different temperatures. Drying kinetics were obtained and fitted to several thin layer mathematical models. Powders were characterized in terms of physicochemical and antioxidant properties, as well as of probiotic potential. Fermentation promoted faster drying rates and increased phenols and flavonoids retention. Increasing drying temperature shortened the process and increased powders’ antioxidant activity. Among the models applied, Page resulted in the best fit for all samples. Microbial survival was favoured by lower drying temperatures (air-drying at 50 °C and freeze-drying). Fermentation and drying conditions were proved to determine both drying behaviour and powders’ properties.

## 1. Introduction

Food waste is an issue of major global concern due to the associated social, economic and environmental consequences [1]. In vegetable distribution channels, around 30% of the total production becomes waste or byproduct [2,3]. Efforts to combat this are emphasised in the 2030 Sustainable Development Goals, which prioritise continuous and sustainable improvement of the food system, while meeting the food growing demand and driving climate action [1]. In this context, it is necessary to search for new recovery strategies that contribute to circular economy of food waste. Specifically, for plant waste, residues are rich in nutrients and bioactive compounds such as polysaccharides, proteins, dietary fibres and antioxidant compounds among others [4,5], making them interest materials for the development of functional ingredients.

Broccoli (*Brassica oleracea* L. var. *italica*) is one of the most widely consumed vegetables of the Brassicaecea family, both in fresh and processed form [6]. Several epidemiological studies have identified an inverse correlation between broccoli consumption and the risk of diseases such as cancer, cardiovascular diseases, neurological diseases or diabetes [7,8]. Bioactive compounds in broccoli include isothiocyanates, glucosinolates, phenols, flavonoids or vitamins [9] that are involved in antioxidant activity, enzyme regulation or control of apoptosis and cell cycles [6]. Currently, around 600,000 tonnes of broccoli are produced annually in Spain [10], of which only 10–15% are intended for human consumption [11]. The parts of broccoli consumed as food are mostly florets, which constitute only 15% of the total plant weight. Leaves and stems, which, respectively, represent approximately 47% and 38% of the total weight, are usually discarded [12]. 

In the last years, manufacturing powders from vegetable wastes has arisen as a useful and effective valorisation alternative to take advantage of their nutritional potential, since powders are stable and versatile products that can be consumed directly or as an ingredient in food formulation. The combination of drying and milling is an economical and environmentally friendly alternative for the integral valorisation of these wastes [13]. Among the dehydration techniques, hot-air drying (HAD) is the most widely used; however, drying conditions (temperature and relative humidity of the air, load density, type of contact between the air and the material to be dried, etc.) must be properly selected to minimise thermal degradation of bioactive compounds and to preserve the nutritional value of foods [6]. HAD is a complex process involving simultaneous mass and heat transfer [14]. Modelling of the drying process is required to predict optimum drying parameters and the drying behaviour of the material [14,15]. Several mathematical models are proposed to describe the drying kinetics of food, being the thin-layer drying models (e.g., Page, Henderson–Pabis, Lewis and Linear) the most widely used to fit the drying curves and estimate the drying time [14,16]. On the other hand, freeze-drying (FD) characterizes by maintaining the appearance, shape, taste and biological activity of foods [7], but it is an expensive technique with a more difficult industrial implementation. 

Pre-drying treatments also play a decisive role on the functional properties of dehydrated products. Disruption intensity directly affects the drying time and the exposure to the airstream, and thus the content of biologically active compounds [17,18,19]. Furthermore, disruption may facilitate the extraction of bioactive compounds and increases their bioavailability, as reported for several plant tissues [9,20]. Another pretreatment is fermentation, an ancient technique considered one of the most effective ways of preserving foods due to the formation of organic acids, alcohols, bacteriocins and other antimicrobial products [21,22]. Fermented matrices may result in improved nutritional properties and high applicability in the food industry as food supplements, seasonings or probiotic foods [23,24], formulation of baby food [25] or fortified cereal-based products [26]. In recent years there has been a growing interest in studying the influence of fermentation on foodstuffs and its benefits for human health [27], as well as on the improvement of functional, sensory and physicochemical properties [28,29]. Among the microorganisms involved in fermentation processes, lactic acid bacteria (LAB) belonging to the genera *Enterococcus*, *Streptococcus*, *Leuconostoc*, *Lactobacillus* and *Pediococcus* stand out [30]. These are microorganisms of industrial importance since they are commonly consumed in foods such as yoghurt, cheese, beer, wine and meat products [31]. According to the WHO, probiotics are “live microorganisms that, when consumed in adequate amounts (10^8^–10^9^ CFU/day), bestow health benefits to the host”. To designate a product as a probiotic requires not only to contain the microorganism in adequate concentration at the time of consumption (>10^6^–10^7^ CFU/g), but also evidence that the strain is safe and beneficial to health. 

During fermentation, LAB bacteria produce primary and secondary metabolites, such as organic acids, CO_2_, hydrogen peroxide and antimicrobial peptides that inhibit the growth of pathogenic microorganisms [32,33]. Acid-lactic fermentation also improves the nutritional and health attributes of foods by bioconverting phytochemical compounds, such as polyphenols, into more bioactive and bioaccessible forms [34,35]. On the other hand, fermentation modifies the raw material structure, loosening the cell wall and increasing the size of tissue pores [36], which would determine the fermented material behaviour during subsequent drying. *L. plantarum* has been reported as the most widely used for fermenting plant substrates and it is a versatile strain that enhances fermented products properties [37,38]. 

In this context, the aim of the present work was to evaluate the impact of pretreatments (disruption and fermentation with *Lactiplantibacillus plantarum*) on the drying behaviour of broccoli stems and the properties of the resulting powders, including physicochemical, antioxidant and microbial survival.

## 2. Materials and Methods

### 2.1. Raw Material and Preconditioning

Broccoli was purchased from a local supermarket in Valencia (Valencia, Spain) and the stems were manually separated with a knife. To reduce microbial load and avoid possible substrate competition during fermentation, fresh broccoli stems were washed by immersion in sodium hypochlorite (200 ppm) solution in water for 5 min. After disinfection, the stems were disrupted in a Thermomix^®^ food processor (Vorwerk, Madrid, Spain) for 30 s at 10,000 rpm (ground samples, G) or 10 s at 10,000 rpm (chopped samples, C). Disruption conditions were established according to previous research [17], so that the maximum particle size was 5 or 10 mm for ground or chopped samples, respectively. The disrupted tissue was distributed among 250 mL sterile glass containers with twist-off closure (200 g of sample per jar) and blanched by immersion in a water bath until reaching 72 °C for 1 min in the pot centre. Samples were then cooled with running water to 40 °C. 

### 2.2. Fermentation with Lactiplantibacillus Plantarum

Disrupted and blanched broccoli stems underwent fermentation with *Lactiplantibacillus plantarum* CECT 749 (Colección Española de Cultivos Tipo, Valencia, Spain). Selection of the microorganism was based on its potential probiotic effect and its role in spontaneous fermentation of fruits and vegetables [37]. Recovery of the freeze-dried strain was carried out in Man, Rogosa and Sharpe (MRS) broth (Scharlab, Barcelona, Spain) at 37 °C for 24 h. The starter culture obtained contained 8.6 ± 0.3 log CFU/mL (measured by plate count) and inoculation was performed by adding 2 mL of it to each jar containing the blanched and cooled plant material. Fermentation tests were carried out in duplicate in an incubation camera (Incudigit-TFT, J.P.Selecta, Barcelona, Spain) at 37 °C for 96 h, and microbial counting was performed at 0, 7, 24, 48, 72 and 96 h. Results of these pretests are given as Appendix A and allowed to select the disruption pretreatment and fermentation time, which were set at ground broccoli stems fermented during 24 h. 

### 2.3. Dehydration Conditions and Powders Obtaining

Chopped and ground broccoli stems, the latter fermented and non-fermented, were either freeze-dried (FD) or hot-air dried (HAD) at 50, 60 or 70 °C until lowering water activity (a_w_) to 0.3 to ensure powder stability [39]. Before FD, samples (ground) were deep-frozen at −40 °C for 24 h in a CVN-40/105 freezer (Matek, Barcelona, Spain) prior to sublimation of the frozen water for 24 h under freezing conditions (−45 °C) and sub-atmospheric pressure (P = 0.1 mbar), followed by nonfreezable water desorption in a LyoQuest-55 laboratory freeze-dryer (Telstar, Terrassa, Spain). For HAD, a convective CLW 750 TOP+ transverse flow tray dryer (Pol-Eko-Aparatura SPJ, Katowice, Poland) with air at 2 m/s and at 50, 60 or 70 °C was used. Samples (ground or chopped) were distributed in 1 cm-thick layers on perforated dryer trays with a load of 200 g of residue per tray. After dehydration, dried samples were milled using a Thermomix^®^ food processor (Vorwerk, Madrid, Spain) at 10,000 rpm for 2 min (at 30 s intervals) to obtain a fine powder [17]. Powders were stored in glass jars in a light-free environment until analysis. 

### 2.4. Drying and Drying Rate Curves—Modelling of the Thin-Layer Drying Curves

Weight variation was registered every 30 min during the first 6 h and every hour until 24 h of drying treatment. Water activity was measured every hour until a target value below 0.3 was obtained. This procedure was carried out as explained elsewhere [17,18] and allowed to obtain the drying curves and drying rate curves of broccoli stems as a function of the different pretreatments applied (fermented/non-fermented, chopped/ground). This experimental procedure also allowed to determine the time needed to reach the target a_w_ value for each pretreated residue, which were: 15 h for non-fermented chopped or ground broccoli residue dried at 50 °C (NF_C_HAD50 and NF_G_HAD50), 12 h for fermented ground broccoli residue (F_G_HAD50), 10 h for non-fermented chopped or ground broccoli residue dried at 60 °C (NF_C_HAD60 and NF_G_HAD60), 7 h for fermented ground broccoli residue dried at 60 °C (F_G_HAD60) and non-fermented chopped broccoli residue dried at 70 °C (NF_C_HAD70), 6 h for non-fermented ground broccoli residue dried at 70 °C (NF_G_HAD70) and 5 h for fermented ground broccoli residue dried at 70 °C (F_G_HAD70).

Modelling the drying curves is necessary to understand the drying characteristics of disrupted broccoli stems. For this purpose, experimental data of the drying process of broccoli samples corresponding to the falling drying rate period (FDRP) were fitted to 5 commonly used thin-layer drying models listed in Table 1 [40].

The selection included theoretical models (such as the diffusional model) that are based on the general theory of heat and mass transfer laws, but are more difficult to apply; semi-empirical models (such as the Lewis model, the Henderson and Pabis model or the Page model) derived from the Fick’s second law of diffusion or the Newton’s law of cooling; and empirical models (such as the linear one) formulated from the direct relationship between the moisture content and the drying time, and whose characteristic parameters may have no physical meaning [41]. Semi-empirical models selected included the Lewis model, which is one of the simplest models describing moisture movement for food products, although it is not exactly the most accurate; the Page model, which is an empirical modification of the Lewis model that usually gives better results for the prediction of moisture loss; and the Henderson and Pabis model, which effectively predicts the drying rate at the beginning of the drying process, but appears sometimes to be less efficient for the last stages of the process [42]. In all these models, MR represents the dimensionless moisture ratio or the reduced driving force (Equation (1)).
(1)$$\mathrm{MR}=\frac{X_{t}^{w}-X_{\mathrm{eq}}^{w}}{X_{c}^{w}{-X}_{\mathrm{eq}}^{w}}$$
where:

$X_{t}^{w}$ is the moisture content of the product on a dry basis at a given time; $X_{c}^{w}$ is the critical moisture content of the product expressed on a dry basis, which coincides with the moisture content at the initial time of the FDRP; and $X_{\mathrm{eq}}^{w}$ represents the equilibrium moisture content of the product on a dry basis.

The goodness of fit of the selected kinetic models to the experimental data was evaluated with the sum squared error (SSE) and the correlation coefficient (R^2^).

### 2.5. Analytical Determinations

#### 2.5.1. Physicochemical Properties

Water activity (a_w_) was obtained with an Aqualab^®^ 4TE dew point hygrometer (Decagon Devices Inc., Pullman, Washington, DC, USA) at 25 °C. Moisture content (x_w_) was determined following the gravimetric double weighing procedure AOAC 934.06 [43], based on the removal of the water present in a known amount of sample by drying in a vacuum oven (Vaciotem-T, JP Selecta, Barcelona, Spain) (P = 10 mm Hg) at 60 °C until reaching constant weight. Total soluble solids content (x_ss_) was calculated from the measurement of the Brix degrees obtained at 20 °C by a thermostated Abbe refractometer NAR-3T (Atago, Tokyo, Japan). In powders, measurements were performed in an aqueous extract 1:10 (*w*/*v*) ratio. pH was measured with a digital pH-meter S20 SevenEasyTM (Mettler-Toledo Inlab, Columbus, OH, USA), previously calibrated with the pertinent buffer solutions at pH 7 and 4. Particle size distribution was determined in a Mastersizer 2000 laser diffraction equipment (Malvern Panalytical Ltd., Malvern, UK) coupled to a unit Hydro 2000. The analysis was carried out with a particle absorption index at 0.1 and refraction indexes of 1.52 and 1.33 for the sample and the dispersed phase (deionized water), respectively. Results of particle size measurements were given in terms of equivalent volume mean diameter D [3,4], surface area mean diameter D [2,3] and distribution percentiles d_10_, d_50_ and d_90_.

#### 2.5.2. Antioxidant Properties

Antioxidant compounds were extracted by mixing the samples with an 80% (*v*/*v*) methanol/bidistilled water in a 4:10 (*w*/*v*) or 1:10 ratio (*w*/*v*) for raw material or powders, respectively. The mixture was stirred in darkness for 1 h in a horizontal stirrer (Magna Equipments S. L., model ANC10, Barcelona, Spain), and then centrifuged at 10,000 rpm for 5 min (Eppendorf Centrifuge 5804/5804R, Madrid, Spain). When necessary, dilutions of extracts were carried out for the analysis below.

Total phenolic content was determined by the Folin–Ciocalteu spectrophotometric method [44,45]. 0.125 mL of the extract were mixed with 0.5 mL of bidistilled water and 0.125 mL of Folin–Ciocalteu (Scharlab S.L., Barcelona, Spain) reagent. After 6 min of reaction in darkness, 1.25 mL of 7% (*w*/*v*) sodium carbonate solution and 1 mL of bidistilled water were added. This preparation was kept in darkness for 90 min and the absorbance was measured at 760 nm in a Helios Zeta UV/Vis spectrophotometer (Thermo Fisher Scientific Inc., Waltham, MA, USA). Results were expressed as mg of Gallic Acid Equivalents (GAE) per g of dry matter.

Total flavonoid content was determined using the modified aluminium chloride colorimetric method [46]. 1.5 mL of the extract was mixed with 1.5 mL of a 2% (*w*/*v*) aluminium chloride solution (Thermo Fisher Scientific Inc., Waltham, MA, USA). After 10 min of reaction in darkness, absorbance was measured at 368 nm with a Helios Zeta UV/Vis spectrophotometer (Thermo Fisher Scientific Inc., Waltham, MA, USA). Results were expressed as mg of Quercetin Equivalents (QE) per g of dry matter.

Antioxidant activity was evaluated following the DPPH (1,1 diphenyl-2-picryl hydrazyl) and ABTS (2,20-azobis-3-ethyl benzothiazolin-6-sulphonic acid) methods. For the DPPH method [47], 0.1 mL of the extract and 2.9 mL of a 0.1 mM DPPH (Merck KGaA and affiliates, Darmstadt, Germany) solution in methanol were mixed. After 60 min of reaction in darkness, absorbance was measured at 575 nm. Results were expressed as mg of Trolox Equivalent (TE) per g of dry matter. The ABTS method was applied according to the procedure described by Re et al. [48]. 0.1 mL of the extract was mixed with 2.9 mL of ABTS^+^ (VWR International LLC, Radnor, PA, USA) solution in phosphate buffer with an absorbance of 0.70 ± 0.02 at 734 nm. After 7 min of reaction the absorbance was measured at 734 nm. Results were expressed as mg of Trolox Equivalent (TE) per g of dry matter.

#### 2.5.3. Microbial Counts

Viable cells of *Lactiplantibacillus plantarum* were estimated by serial dilution from 10^−1^ to 10^−8^ g/L with buffered peptone water (Scharlab, Barcelona, Spain), seeding on MRS agar (Scharlab, Barcelona, Spain) and incubation at 37 °C for 24 h. First dilution (10^−1^ g/L) was obtained by mixing 3 g of solid sample with 27 mL of sterile buffered peptone water in a stomacher bag and homogenizing for 2 min. Finally, colonies present on the plates were counted.

### 2.6. Statistical Analysis

Statistical analysis was carried out with Statgraphics Centurion XVII software (Statpoint Technologies, Warrenton, VA, USA), applying simple and multifactorial analysis of variance (ANOVA) with a confidence level of 95% (*p*-value < 0.05).

## 3. Results and Discussion

### 3.1. Impact of Fermentation on Broccoli Wastes Characteristics

Table 2 summarizes physicochemical and antioxidant properties of the disrupted and blanched broccoli stems before and after fermentation during 24 h with *Lactiplantibacillus plantarum* CECT 749. Moisture content, water activity and soluble solid content were in the range of the reported in previous studies [18,49]. As expected, moisture (x_w_) and water activity (a_w_) values before fermentation were considerably high, thus confirming that the broccoli residue represents a good environment for bacterial growth. No statistically significant differences were obtained between fermented and non-fermented broccoli stems regarding moisture (x_w_), water activity (a_w_) and total soluble solids content (x_ss_), although other authors have reported a decrease in total solutes during fermentation with *L. plantarum* [50]. Total phenols content was similar to values previously reported for raw broccoli stems (3.8 ± 0.2 mg GAE/g_dm_) [18], but flavonoid content presented lower values than in other studies (4.6 ± 0.7 mg QE/g_dm_ or 2.4 ± 0.1 mg QE/g_dm_) [18,51]. Likewise, the antioxidant capacities obtained by the DPPH and ABTS methods were lower than the values reported in other studies [18,52]. These differences might be attributed to different broccoli varieties, culture conditions or other agronomic factors [52].

Results evidenced statistically significant differences between fermented and non-fermented samples, fermentation generally having a positive impact on antioxidant characteristics, except for phenols, where no effect was observed. In Tkacz et al. [50] fermentation by using *L. plantarum* enhanced flavonols and antioxidant activity of sea buckthorn berries mixed with apple juices. Lactic acid bacteria have been reported to increase antioxidant properties in fermented plant-based foods due to a decrease in pH and enzyme hydrolysis during fermentation [53]. Fermentation mainly improves the release of antioxidant compounds via microbial hydrolysis; nevertheless, fermentation can also induce structural breakdown of plant cell wall liberating and/or synthesising new antioxidant compounds, as well as phytochemicals’ structural changes enhancing or modifying antioxidant activity [54]. It has been reported that fermentation can have a positive or negative impact on the release and production of antioxidant compounds, depending on the microorganism, fermentation conditions or vegetable matrix [53,54,55].

### 3.2. Drying Curves and Drying Rate Curves

Drying curves and drying rate curves obtained for the different treatments assayed are presented in Figure 1. Drying curves represent the evolution over time of the quotient between the moisture at each instant of the process ($X_{t}^{w}$) and the initial moisture ($X_{0}^{w}$), both expressed on a dry matter (kg w/kg db). Drying rate curves represent the rate at which moisture content decreases along drying with respect to the amount of water present in the sample. 

As observed, the drying time needed to reach a similar moisture content decreased as drying air temperature increased. Disruption did not have a significant impact on drying kinetics; in contrast, fermentation also shortened the drying process. This effect of fermentation on drying behaviour could be attributed to microorganisms’ action, i.e., microbial metabolism and enzyme production, such as cellulases and glycosidases which degrade the cell wall and may reduce tissue resistance to water transport [56], or produce structural changes which increase the amount of free or unbound water.

The drying curves obtained were used to determine the moisture content of the samples in equilibrium with the airstream ($X_{\mathrm{eq}}^{w}$). No significant differences were found between ground and chopped samples, whereas fermentation and drying temperature did imply a reduction in the equilibrium values so that these decreased from 0.074 to 0.068 kg w/kg db for samples air-dried at 50 °C, from 0.027 to 0.018 kg w/kg db for samples air-dried at 60 °C and from 0.018 to 0.011 kg w/kg db for samples air-dried at 70 °C.

In general, drying rate curves (Figure 1) exhibited an initial period in which drying was predominantly controlled by the rate at which water is removed from the surface of the product, which is determined by air conditions and product a_w_. This is known as the constant drying rate period (CDRP), during which free or loosely bound water is removed and the risk of sample damage by exposure to high temperatures is low, since the temperature of sample is close to the wet bulb temperature of the drying air [18,57,58].

As drying proceeds, replacement of water evaporated from the surface of the product by water from the inside becomes more difficult, so that dry regions start to appear on the product surface. The drying rate decreases progressively so this period is known as the falling drying rate period (FDRP), during which the risk of thermolabile compounds degradation is higher as part of the energy is used to heat up the sample, increasing its temperature. The critical time (t_c_) and corresponding critical moisture content ($X_{c}^{w}$) defines the transition from the CDRP to the FDRP. Average drying rate during the CDRP and critical moisture content for the different conditions assayed are summarized in Table 3.

In general terms, the duration of the CDRP decreased with the drying temperature and when fermentation was applied as a pretreatment. Indeed, in some cases the drying seemed to directly enter a FDRP, mainly when higher drying temperatures were combined with fermentation: i.e., higher a_w_ gradients which implied higher drying rates. Nevertheless, the initial response was approximated to a CDRP in all cases for practical purposes. This response has been reported elsewhere and is a consequence of the higher initial drying rates promoting case-hardening phenomena which affects water diffusion in the tissue matrix by increasing internal resistance to water transfer [18,57]. Regarding average drying rate during the CDRP, no significant differences (*p*-value < 0.05) were found between drying rate values in the CDRP for ground and chopped samples dried at a given temperature. However, the critical moisture content was significantly affected by the intensity of the previous disruption, especially at 60 and 70 °C.

### 3.3. Modelling of the Drying of Disrupted Broccoli Stems in Thin Layers

Experimental data corresponding to the FDRP drying kinetics were fitted to the mathematical models listed in the Materials and Methods section, and corresponding parameters are given in Table 4, including drying model coefficients and the comparison criteria R^2^ and SSE used to evaluate goodness of fit.

Statistically significant differences (*p*-value < 0.05) of k, *n* and a parameters were observed among samples in all models tested. Generally, kinetic parameters evidenced higher drying rates and mass transfer coefficients (k) in fermented samples and when temperature increased from 50 to 70 °C, in agreement with previous reports [40,59] since at higher temperatures the energy available for water mobility and evaporation rises, thus increasing the drying rate [59]. On the other hand, results did not prove a significant effect of the disruption treatment. 

The proposed models fitted the data adequately, showing R^2^ values generally higher than 0.90 and low SSE values. Among models, Page resulted in the best fit for all samples, with the highest R^2^ values (R^2^ ≥ 0.98) and the lowest SSE values varying from 0.0007 to 0.0089. The Page model has shown good applicability for both dehydration and rehydration processes of several food products, subjected to different pretreatments and drying methods [15,60]. In the present study, k values, related to the diffusion coefficient and the geometry of the sample [61], and *n* values, related to the type of diffusion and the microstructure of the food [61], are in the range of those obtained by Md Salim et al. [14] in drying of broccoli stems cut into 6 mm thick slices at 40 °C (k = 0.12 h^−1^ and n = 1.205), 50 °C (k = 0.138 h^−1^ and n = 1.230) and 60 °C (k = 0.144 h^−1^ and n = 1.288). k increased with the temperature and in fermented samples. Although the intensity of the disruption pretreatment did not significantly affect the value of the parameter k in the samples dried at 50 and 60 °C, at 70 °C grinding resulted in significantly higher k values than chopping and closer to values obtained for fermented samples. Since the values obtained for *n* are higher than 1, it can be stated that drying broccoli stems under the conditions studied is a super-diffusion process [61]. For a similar temperature, the value of *n* was not significantly affected by the intensity of previous disruption, but increased significantly with fermentation.

To confirm the goodness of fit of Page’s model, Figure 2 plots the predicted MR values against experimental ones. The closeness of the plotted data to the straight line represents the good correlation between calculated and experimental results. Likewise, the drying curves predicted with the Page model and experimental ones are also plotted.

Several empirical equations are usually used to evaluate and model the drying kinetics of food, such as Page, Lewis, Henderson–Pabis and Linear models studied in this work, among others. All of these models are ultimately governed by the diffusional model of Fick’s second law, which is the most widely studied theoretical model in thin-layer drying of different food products [15,62]. Diffusivities obtained from the first term of the analytical solution to Fick’s second law of diffusion proposed by Crank [63] for an infinite layer geometry and long treatment times, were of the same order as those obtained by other authors for similar products, e.g., effective diffusivity of fresh broccoli florets ranged from 2.82 × 10^−10^ to 2.00 × 10^−9^ m^2^/s in Mahn et al. [64] or even increased to an average value of 4.9 × 10^−9^ m^2^/s in Reyes et al. [65], and in the case of blanched broccoli taking values between 1.99 × 10^−8^ and 3.56 × 10^−8^ m^2^/s [16]. The statistical analysis confirmed that increasing the air temperature or applying a fermentation step prior to drying significantly increased the value of the effective water diffusivity. Higher effective diffusivity values imply faster drying processes. Shortening the drying process could be positive for preserving the probiotic properties of the powders; however, the negative impact of temperature must also be evaluated to determine the optimal process conditions to ensure probiotic viability in the final product [59]. No significant differences were found between the effective water diffusivity values of chopped and ground samples dried at the same temperature.

### 3.4. Characterization of Broccoli Stem Powdered Products

#### 3.4.1. Moisture Content, Water Activity, Antioxidant Properties and Particle Size Characteristics

Table 5 shows the values of moisture content (x_w_), water activity (a_w_), total phenols and flavonoids content, as well as the antioxidant activity measured by DPPH and ABTS methods of the different powders. Results indicate that drying significantly decreased the moisture and a_w_ values of the raw material (Table 2). The a_w_ indicates the availability of water to participate in reactions responsible for food spoilage, so values below 0.3, as obtained, guarantee the powder’s stability [39].

In all cases, the antioxidant properties of the powders were improved as compared to non-processed stems (Table 2). Powders fermented with *Lactiplantibacillus plantarum* CECT 749 generally presented higher total phenols and flavonoids content, but one of the lowest antioxidant capacities measured by both DPPH and ABTS, particularly when drying at 60 or 70 °C. These results are consistent with other studies, such as that of Cai et al. [34] in which fermentation significantly increased total phenol content, but to a lesser extent the antioxidant capacity of broccoli puree. This could be explained by microbial enzymes which degrade cell wall polysaccharides, mainly cellulases and glycosidases, favouring the release of phenolic compounds [56,66], as well as to the fermentative microorganism production of new antioxidant compounds [67] and the transformation of some antioxidant compounds into others with higher activity [56]. Moreover, fermented powders required less drying time, which decreased the degradation of phenolic compounds by oxidation.

Statistically significant differences (*p*-value < 0.05) were found among powders regarding total phenols and flavonoids. HAD powders, particularly at 70 °C, presented the highest values. It has been reported that higher drying temperatures might increase phenolic content by contributing to the formation of new antioxidant compounds due to Maillard reactions [68] together with the reduction in certain enzymes’ activity capable of degrading phenolic compounds [69]. Moreover, shorter exposure time to drying conditions at 70 °C could reduce phenols’ degradation [18,20]. Among HAD samples, differences between ground and chopped samples could be attributed to the different duration of the CDRP. Considering that during this period the temperature of the sample remains closer to the saturation temperature of the air, the exposure time to elevated temperatures in chopped samples was shorter, so their total phenol and flavonoid contents were generally higher. This trend had also been observed in a previous study [18]. FD ground broccoli stems exhibited better phenol and flavonoid content compared to chopped, which could be attributed to an intensified structure disruption while freezing. Antioxidant capacity by the ABTS method resulted in higher values than DPPH, which may be due to a greater affinity of the antioxidant compounds present in the broccoli stem with this free radical. HAD temperature did not have a significant effect on antioxidant capacity. Grinding before HAD significantly reduced (*p*-value < 0.05) the ability to inhibit the DPPH and ABTS radicals; in contrast, grinding prior to FD resulted in a higher capacity to inhibit the ABTS radical and a lower capacity to inhibit the DPPH radical. To explain this, it should be considered that antioxidant capacity could be affected by the presence of other compounds with antioxidant activity which have not been quantified in this study, but that could have a different behaviour under the processing conditions used.

Figure 3 shows the particle size distribution and characteristic parameters for the different powders obtained. Fermentation had a different impact on particle size depending on the drying technique and conditions applied. In FD samples, fermented ones exhibited the smallest particle size; however, coarser particles were obtained for HAD fermented samples as compared to non-fermented ones, at the lowest temperature assayed. On the one hand, the presence of viable microorganism might modify particle size by increasing it; on the other, the fermentative action could imply a more intense disruption of the tissue, thus having an impact on the structure, which could respond differently to drying yielding different particle characteristics after milling. In general, lowering drying HAD temperature implied wider spans and coarser particles, whereas FD gave rise to finer powders, a result which could be attributed to the more porous and fragile structure of FD products [39,70]. Grinding yielded finer particles when HAD; the opposite response was found in FD powders. 

#### 3.4.2. Potential Probiotic Properties

Microbial counts in fermented broccoli stems and resulting powders are presented in Figure 4. Microbial survival in FD and HAD50 powders was high enough to consider the powders as probiotic ingredients, since these contained more than 10^7^ CFU per g of product [71,72]. Air-drying at higher temperatures implied a reduced probiotic content. Similar trends were observed in a previous study on persimmon wastes fermented with *L. salivarius* [13], in which microbial counts in FD powders were 10^7^ CFU/g whereas HAD affected microbial viability in a greater extent. Other authors have reported higher microbial survival (>10^8^ CFU/g) in freeze-dried apple snacks enriched with *L. plantarum* [73]. Low temperature HAD, particularly 40 °C, has been demonstrated to retain 10^6^ CFU/g of *L. casei* in dried murta berries [15]. Ultrasound-assisted HAD has been reported to improve *L. casei* viability in apple snacks due to drying shortening [59].

## 4. Conclusions

This research has demonstrated that fermentation used as a pretreatment has a significant impact on broccoli stems drying response, given that it has promoted faster initial drying rates and has allowed to shorten the drying process. Different mathematical models have been adjusted to the experimental results and their characteristic parameters have confirmed the effect of fermentation applied prior to drying and drying temperature on drying rates. In contrast, intensity of previous disruption has not been proved to have a significant effect, under the conditions assayed. Conditions which shortened the drying treatment (fermentation and temperature) had a positive impact on the antioxidant properties of the powdered products obtained due to both, the biochemical and structural changes undergone and the reduction in the time of exposure to drying conditions.

The results obtained suggest that fermentation is a simple and economic method which may result in an increased nutritional value and reduced drying times, thus resulting in energy savings and reducing processing costs. However, as for probiotic survival, reducing the impact of drying implies using more expensive and difficult to implement techniques such as freeze-drying, or lowering the temperature of the air, which implies lengthening the process.

Different processing alternatives have been applied to transform broccoli stem residues into stable powdered products, which could be proposed to be used as functional ingredients to fortify foods, according to their antioxidant and probiotic properties.

## Figures and Tables

**Figure 1 foods-12-03526-f001:**
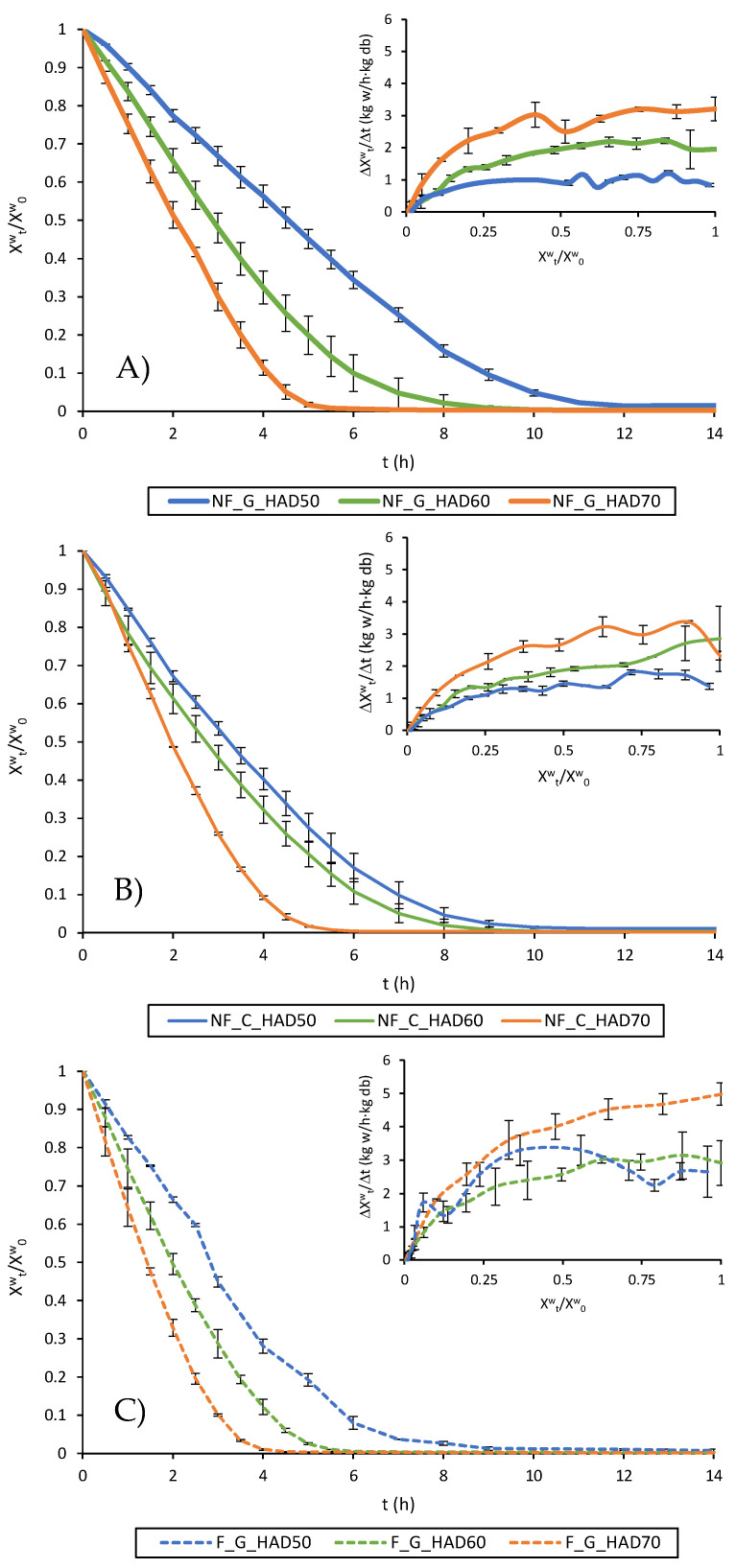
Drying and drying rate curves of: (**A**) non-fermented and chopped (NF_C); (**B**) non-fermented and ground (NF_G); and (**C**) fermented and ground (F_G) broccoli stems hot-air dried at 50 (HAD50), 60 (HAD60) and 70 °C (HAD70).

**Figure 2 foods-12-03526-f002:**
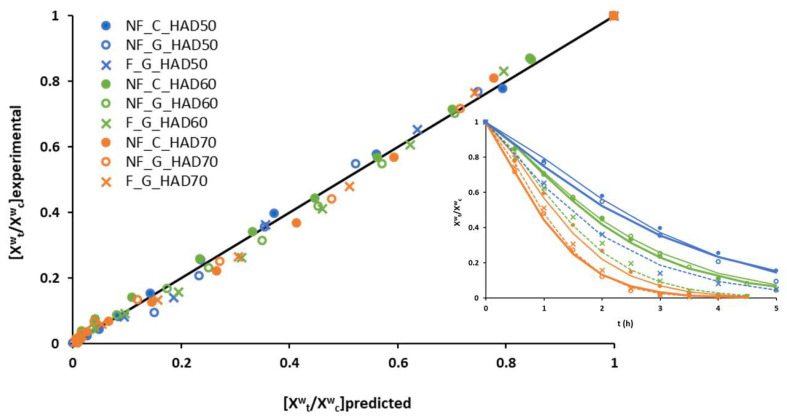
Goodness of fit of Page’s model: comparison between experimental and predicted reduced moisture values and comparison between experimental (markers) and predicted (lines) drying curves.

**Figure 3 foods-12-03526-f003:**
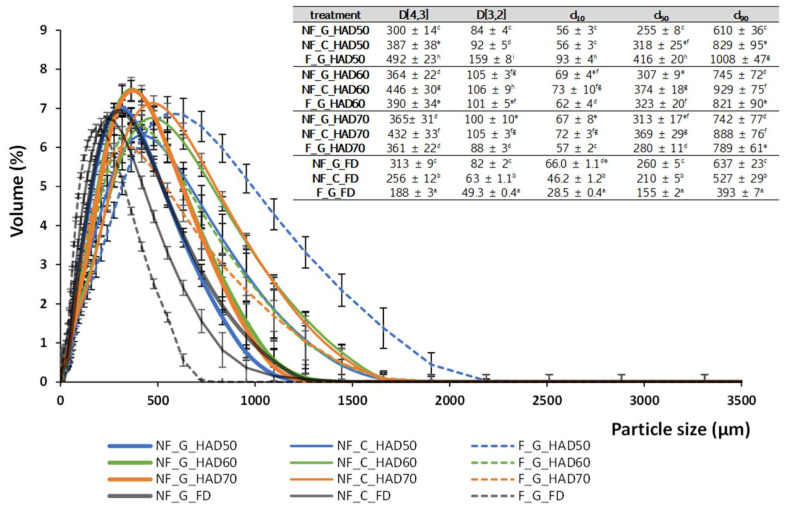
Particle size distributions and particle size characteristic parameters of the broccoli stem powders (equivalent volume diameter D[3,4], surface area mean diameter D[2,3], percentiles d_10_, d_50_ and d_90_) Error bars are the standard deviation of five replicates. HAD: hot air-drying at 50, 60 or 70 °C, FD: freeze-drying; C: chopped, G: ground; NF: non-fermented, F: fermented. Mean ± standard deviation of five replicates. ^a–i^ Different superscript letters in the same column indicate statistically significant differences at the 95% confidence level (*p*-value < 0.05).

**Figure 4 foods-12-03526-f004:**
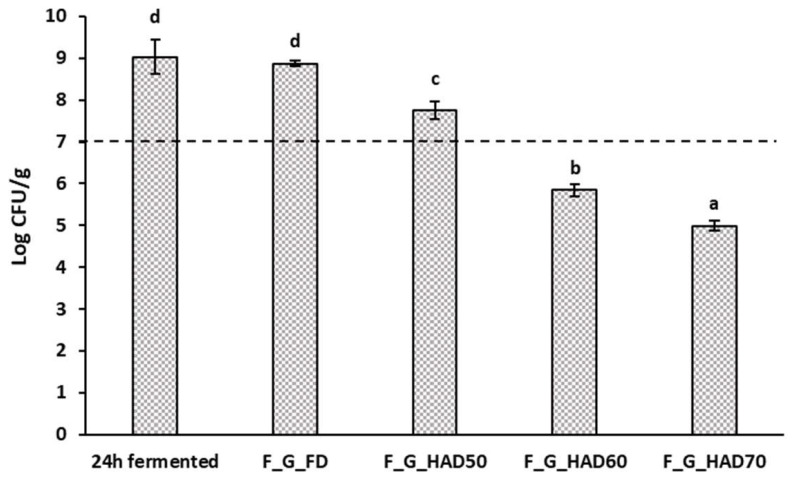
Viable counts (*L. plantarum*) in powders obtained by pre-fermentation of broccoli stems. Error bars represent the standard deviation of four replicates. HAD: hot-air drying at 50, 60 or 70 °C, FD: freeze-drying; C: chopped, G: ground; F: fermented. ^a–d^ Different letters in the same series indicate significant differences at the 95% confidence level (*p*-value < 0.05).

**Table 1 foods-12-03526-t001:** Models used to analyse the drying kinetics of broccoli stems during the falling drying rate period.

Model	Equation
Lewis	MR = exp(−k · t)
Henderson and Pabis	MR = a · exp(−k · t)
Linear	MR = −k · t + a
Page	MR = exp(−k · t^n^)
Diffusional	$\mathrm{MR}=\frac{8}{\pi^{2}}\cdot \exp\left(-\frac{D_{\mathrm{ef}}\;\cdot \;\pi^{2}\;\cdot \;t}{4\;\cdot \;l^{2}}\right)$

where a and n are the constants of the kinetic models, k is the rate coefficient, D_ef_ is the effective or apparent diffusivity and l is the half thickness.

**Table 2 foods-12-03526-t002:** Moisture content (x_w_), water activity (a_w_), soluble solids content (x_ss_) and antioxidant properties of non-fermented and fermented broccoli stems. Mean ± standard deviation of triplicates from two replicas.

Property	Non-Fermented	Fermented
x_w_ (%)	90.5 ± 1.0 ^a^	92 ± 2 ^a^
a_w_	0.989 ± 0.005 ^a^	0.991 ± 0.003 ^a^
x_ss_ (g/g)	0.696 ± 0.062 ^a^	0.615 ± 0.035 ^a^
Total Phenol Content (mg GAE/g_dm_)	3.62 ± 0.17 ^a^	3.7 ± 0.4 ^a^
Total Flavonoid Content (mg QE/g_dm_)	0.75 ± 0.15 ^a^	1.2 ± 0.3 ^b^
DPPH (mg TE/g_dm_)	0.46 ± 0.16 ^a^	1.1 ± 0.3 ^b^
ABTS (mg TE/g_dm_)	3.66 ± 0.06 ^a^	5.46 ± 0.11 ^b^

^a,b^ Different superscripts in the same property indicate statistically significant differences at 95% confidence level (*p*-value < 0.05).

**Table 3 foods-12-03526-t003:** Values of critical time (t_c_), critical moisture content ($X_{c}^{w}$) and drying rate in the constant drying rate period (rate_CDRP_) of disrupted broccoli stems as a function of pretreatment (G: ground; C: chopped; F: fermented; NF: non-fermented) and temperature of drying air (HAD50: 50 °C; HAD60: 60 °C; HAD70: 70 °C). Mean ± standard deviation of duplicates from two replicas.

	t_c_ (h)	$X_{c}^{w}$ (kg w/kg db)	Rate_CDRP_ (kg w/h·kg db)
**NF_G_HAD50**	7	4.1 ± 0.2 ^a^	0.97 ± 0.02 ^a^
**NF_C_HAD50**	7	4.4 ± 0.3 ^a^	1.10 ± 0.07 ^a^
**F_G_HAD50**	3	5.5 ± 0.6 ^b^	2.6 ± 0.4 ^cd^
**NF_G_HAD60**	2.5	6.3 ± 0.4 ^c^	2.13 ± 0.12 ^b^
**NF_C_HAD60**	3	5.36 ± 0.08 ^b^	2.2 ± 0.3 ^bc^
**F_G_HAD60**	1.5	6.7 ± 0.2 ^c^	3.0 ± 0.3 ^d^
**NF_G_HAD70**	2.5	4.6 ± 0.4 ^a^	2.99 ± 0.12 ^d^
**NF_C_HAD70**	1.5	6.44 ± 0.15 ^c^	2.98 ± 0.03 ^d^
**F_G_HAD70**	1	7.6 ± 0.3 ^d^	4.73 ± 0.11 ^e^

^a–e^ Different superscripts in the same column indicate statistically significant differences at 95% confidence level (*p*-value < 0.05).

**Table 4 foods-12-03526-t004:** Parameters resulting from fitting mathematical models to broccoli stem drying data during the falling drying rate period (FDRP). SSE: sum squared error. HAD: hot-air drying at 50, 60 or 70 °C; C: chopped, G: ground; F: fermented; NF: non-fermented. Mean ± standard deviation of duplicates from two replicas.

Model	Parameters	NF_G_HAD50	NF_C_HAD50	F_G_HAD50	NF_G_HAD60	NF_C_HAD60	F_G_HAD60	NF_G_HAD70	NF_C_HAD70	F_G_HAD70
**Page**	**k (h^−1^)**	0.29 ± 0.05 ^a^	0.23 ± 0.02 ^a^	0.45 ± 0.04 ^bc^	0.35 ± 0.05 ^ab^	0.36 ± 0.04 ^ab^	0.503 ± 0.005 ^bc^	0.8 ± 0.2 ^d^	0.57 ± 0.02 ^c^	0.73 ± 0.08 ^d^
	** *n* **	1.17 ± 0.08 ^a^	1.32 ± 0.08 ^ab^	1.19 ± 0.13 ^a^	1.30 ± 0.08 ^ab^	1.31 ± 0.06 ^ab^	1.420 ± 0.005 ^b^	1.31 ± 0.10 ^ab^	1.41 ± 0.02 ^b^	1.46 ± 0.05 ^b^
	**SSE**	0.0007	0.0059	0.0075	0.0057	0.0059	0.0089	0.0060	0.0080	0.0069
	**R^2^**	0.983	0.996	0.975	0.992	0.992	0.984	0.979	0.988	0.990
**Diffusional**	**D_ef_ × 10^−9^ (m^2^/s)**	1.34 ± 0.03 ^a^	1.16 ± 0.06 ^a^	1.5 ± 0.13 ^a^	1.5 ± 0.4 ^a^	1.6 ± 0.6 ^a^	2.35 ± 0.02 ^b^	3.0 ± 0.2 ^cd^	2.58 ± 0.07 ^bc^	3.34 ± 0.14 ^d^
	**SSE**	0.0246	0.0388	0.0398	0.0445	0.0492	0.0516	0.0277	0.0497	0.0531
	**R^2^**	0.849	0.938	0.968	0.902	0.901	0.879	0.935	0.903	0.919
**Lewis**	**k (h^−1^)**	0.40 ± 0.04 ^a^	0.44 ± 0.02 ^ab^	0.59 ± 0.05 ^b^	0.6 ± 0.2 ^b^	0.61 ± 0.12 ^b^	0.80 ± 0.13 ^c^	1.14 ± 0.08 ^de^	0.985 ± 0.010 ^cd^	1.26 ± 0.06 ^e^
	**SSE**	0.0254	0.0246	0.0171	0.0294	0.0318	0.0379	0.0266	0.0362	0.0393
	**R^2^**	0.986	0.963	0.981	0.939	0.940	0.911	0.947	0.931	0.941
**Henderson and Pabis**	**k (h^−1^)**	0.40 ± 0.05 ^a^	0.50 ± 0.03 ^ab^	0.61 ± 0.05 ^bc^	0.7 ± 0.2 ^c^	0.71 ± 0.15 ^c^	1.07 ± 0.02 ^d^	1.20 ± 0.04 ^d^	1.155 ± 0.003 ^d^	1.45 ± 0.05 ^e^
	**a**	1.1 ± 0.2 ^a^	1.44 ± 0.05 ^bc^	1.05 ± 0.02 ^a^	1.5 ± 0.2 ^c^	1.5 ± 0.2 ^bc^	1.74 ± 0.06 ^c^	1.2 ± 0.2 ^ab^	1.77 ± 0.07 ^c^	1.72 ± 0.03 ^c^
	**SSE**	0.0023	0.0469	0.0158	0.0497	0.0519	0.0791	0.0265	0.0755	0.0822
	**R^2^**	0.989	0.980	0.982	0.965	0.968	0.946	0.951	0.960	0.957
**Linear**	**k (h^−1^)**	0.183 ± 0.004 ^c^	0.154 ± 0.006 ^a^	0.157 ± 0.004 ^ab^	0.154 ± 0.003 ^a^	0.167 ± 0.002 ^b^	0.224 ± 0.007 ^d^	0.282 ± 0.009 ^e^	0.282 ± 0.002 ^e^	0.334 ± 0.003 ^f^
	**a**	0.953 ± 0.014 ^e^	0.93 ± 0.02 ^de^	0.801 ± 0.004 ^a^	0.83 ± 0.03 ^ab^	1.36 ± 0.02 ^f^	0.860 ± 0.005 ^bc^	0.83 ± 0.05 ^ab^	1.329 ± 0.005 ^f^	0.90 ± 0.02 ^cd^
	**SSE**	0.0016	0.0030	0.0517	0.0315	0.1612	0.0289	0.0625	0.1228	0.0624
	**R^2^**	0.984	0.969	0.836	0.894	0.912	0.925	0.880	0.960	0.949

^a–f^ Different superscript letters in the same row indicate statistically significant differences at the 95% confidence level (*p*-value < 0.05).

**Table 5 foods-12-03526-t005:** Moisture content (x_w_), water activity (a_w_), total phenol content (mg GAE/g_dm_), total flavonoid content (mg QE/g_dm_) and antioxidant activity measured by DPPH and ABTS methods (mg TE/g_dm_) of broccoli stem powders. HAD: hot-air drying at 50, 60 or 70 °C, FD: freeze-drying; C: chopped, G: ground; NF: non-fermented, F: fermented. Mean ± standard deviation of triplicates from two replicas.

Treatment	x_w_ (%)	a_w_	Total Phenol Content (mg GAE/g_dm_)	Total Flavonoid Content (mg QE/g_dm_)	DPPH (mg TE/g_dm_)	ABTS (mg TE/g_dm_)
**NF_G_HAD50**	3.7 ± 0.2 ^f^	0.28 ± 0.04 ^cde^	3.0 ± 0.4 ^a^	2.48 ± 0.13 ^ab^	1.65 ± 0.03 ^a^	12.2 ± 0.4 ^a^
**NF_C_HAD50**	3.3 ± 0.2 ^ef^	0.239 ± 0.008 ^ab^	5.04 ± 0.10 ^cd^	3.24 ± 0.12 ^cd^	2.77 ± 0.14 ^bcd^	13.1 ± 0.3 ^ab^
**F_G_HAD50**	3.92 ± 0.12 ^g^	0.292 ± 0.005 ^e^	6.8 ± 0.6 ^f^	3.5 ± 0.4 ^d^	3.4 ± 0.4 ^de^	19.2 ± 0.7 ^f^
**NF_G_HAD60**	3.3 ± 0.4 ^ef^	0.264 ± 0.014 ^bcd^	3.9 ± 0.5 ^b^	2.9 ± 0.3 ^c^	2.6 ± 0.6 ^bc^	12.1 ± 1.1 ^a^
**NF_C_HAD60**	3.2 ± 0.5 ^def^	0.27 ± 0.03 ^cd^	4.9 ± 0.5 ^c^	4.6 ± 0.2 ^e^	2.6 ± 0.3 ^bc^	26.7 ± 1.5 ^g^
**F_G_HAD60**	2.57 ± 0.04 ^abcd^	0.253 ± 0.004 ^abc^	6.4 ± 0.3 ^ef^	4.5 ± 0.5 ^e^	3.0 ± 0.5 ^bcd^	19.2 ± 0.8 ^f^
**NF_G_HAD70**	2.9 ± 0.6 ^bcde^	0.267 ± 0.012 ^cd^	6.1 ± 0.9 ^e^	4.6 ± 0.4 ^e^	3.1 ± 0.3 ^cd^	14.5 ± 0.9 ^bc^
**NF_C_HAD70**	2.5 ± 0.4 ^abc^	0.26 ± 0.02 ^cd^	6.3 ± 0.9 ^ef^	4.4 ± 0.4 ^e^	3.1 ± 0.4 ^d^	15.3 ± 0.8 ^cd^
**F_G_HAD70**	3.197 ± 0.011 ^cdef^	0.237 ± 0.007 ^a^	8.7 ± 0.3 ^g^	5.7 ± 0.4 ^f^	2.48 ± 0.11 ^b^	13.0 ± 1.1 ^a^
**NF_G_FD**	2.40 ± 0.03 ^ab^	0.262 ± 0.005 ^abcd^	4.2 ± 0.2 ^c^	3.27 ± 0.05 ^d^	3.3 ± 0.4 ^d^	17 ± 3 ^de^
**NF_C_FD**	2.76 ± 0.04 ^abcde^	0.289 ± 0.005 ^de^	3.5 ± 0.2 ^ab^	2.17 ± 0.13 ^a^	3.9 ± 0.4 ^e^	12.8 ± 0.3 ^a^
**F_G_FD**	2.096 ± 0.006 ^a^	0.2490 ± 0.0007 ^abc^	5.88 ± 0.11 ^de^	2.5 ± 0.2 ^b^	2.6 ± 0.2 ^bc^	17.2 ± 0.8 ^e^

^a–g^ Different superscript letters in the same column indicate statistically significant differences at the 95% confidence level (*p*-value < 0.05).

## Data Availability

The data used to support the findings of this study can be made available by the corresponding author upon request.

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
