# Peer review of "Impact of Fermentation Pretreatment on Drying Behaviour and Antioxidant Attributes of Broccoli Waste Powdered Ingredients"

_foods, 2023, doi:10.3390/foods12193526_

Round 1
Reviewer 1 Report
Comments and Suggestions for Authors
The study is interesting and gives a novel insight into the application of fermentation amidst the effort to increase the additional values of food wastes. These are several comments to improve the manuscript:
1. The title could be improved using a better sentence structure to facilitate readers’ understanding. I would suggest the title to be changed to “The effects of pretreatment by fermentation on the drying kinetics and antioxidant activities of powdered broccoli-wastes”.
2. The methodology section is quite difficult to follow. Please use the term “purée” to described broccoli stems that have undergone grinding using Thermomix. L. 106: use purée instead of disruptive tissues. L. 112: please change “disrupted and blanched broccoli steams” into “the purée”. Please check all the methodology regarding this.
3. L. 104-105: please describe the difference between chopped and ground samples.
4. The authors use 5 different drying kinetic models in this study. Please describe the main difference among these models in relation to purée drying. Which method is considered as the most pertinent in this case?
5. Table 2: please discuss why there is no difference in the fermented and non-fermented products regarding total phenolic compounds and flavonoids, but the antioxidant activity (DPPH and ABTS) in the two sample groups differs?
Author Response
Thank you very much for taking the time to review this manuscript. In the document attached you will find the detailed responses to your comments. The corresponding modifications are in track changes in the re-submitted files.

Reviewer 2 Report
Comments and Suggestions for Authors
The manuscript describes a process for the valorisation of broccoli, which was fermented with Lactiplantibacillus plantarum CECT 749 after two pre-treatments. The respective trials did not include inoculation of the strain. The work is topical because it valorises a waste product (broccoli stem residues) rich in antioxidant and probiotic properties, which can be used as an ingredient to fortify various foods. The paper is reasonably well written, clear in form, with an experimental approach appropriate to the purpose. The data were clearly presented and each data point was discussed. I have only minor comments to suggest to the authors to improve the manuscript:
Line 113: Can we learn more about this strain of L. plantarum? Why did the authors use this strain? Has it been used in previous experiments by the same and/or other authors? Even on the same matrix?
Line 113: delete spp. after LAB species name.
Line 113: The strain code should be placed after the species name and not inside the parentheses.
Line 116: Was the same culture medium used for plate counting? Specify.
Line 198: delete underlining.
Line 206: delete underlining.
Line 212: delete underlining.
Lines 229-230: How many replicates have been carried out?
Line 240: Delete the parentheses around the strain code.
Lines 405-406: L. plantarum CECT 749.
Line 463: Did the authors verify the presence of LAB in unfermented broccoli to exclude possible contamination?
Author Response
Thank you very much for taking the time to review this manuscript. In the attached file you will find the detailed responses to your interesting comments. The corresponding revisions are in track changes in the re-submitted files.
